# Sequential Query Encoding
# for Complex Query Answering on Knowledge Graphs

**Jiaxin Bai**[*]                                                                                                    *jbai@connect.ust.hk*
*Department of Computer Science and Engineering*
*Hong Kong University of Science and Technology*

**Tianshi Zheng**[*]                                                                                            *tzhengad@connect.ust.hk*
*Department of Computer Science and Engineering*
*Hong Kong University of Science and Technology*

**Yangqiu Song**                                                                                                    *yqsong@cse.ust.hk*
*Department of Computer Science and Engineering*
*Hong Kong University of Science and Technology*

**Reviewed on OpenReview:** *https://openreview.net/forum?id=ERqGqZzSu5*

## Abstract

Complex Query Answering (CQA) is an important and fundamental task for knowledge graph (KG) reasoning. Query encoding (QE) is proposed as a fast and robust solution to CQA. In the encoding process, most existing QE methods first parse the logical query into an executable computational direct-acyclic graph (DAG), then use neural networks to parameterize the operators, and finally recursively execute these neuralized operators. However, the parameterization-and-execution paradigm may be potentially over-complicated, as it can be structurally simplified by a single neural network encoder. Meanwhile, sequence encoders, like LSTM and Transformer, proved to be effective for encoding semantic graphs in related tasks. Motivated by this, we propose sequential query encoding (SQE) as an alternative to encode queries for CQA. Instead of parameterizing and executing the computational graph, SQE first uses a search-based algorithm to linearize the computational graph to a sequence of tokens and then uses a sequence encoder to compute its vector representation. Then this vector representation is used as a query embedding to retrieve answers from the embedding space according to similarity scores. Despite its simplicity, SQE demonstrates state-of-the-art neural query encoding performance on FB15k, FB15k-237, and NELL on an extended benchmark including twenty-nine types of in-distribution queries. Further experiment shows that SQE also demonstrates comparable knowledge inference capability on out-of-distribution queries, whose query types are not observed during the training process.

## 1 Introduction

Complex query answering (CQA) is a fundamental and important task in knowledge graph (KG) reasoning. CQA can also be used for solving downstream tasks like knowledge-base question answering (KBQA) (Sun et al., 2020). The complex queries on KG are defined in first-order logical form, and they can express complex semantics with the help of logical operators like *conjunction* $\wedge$, *disjunction* $\vee$, and *negation* $\neg$. In Figure 1 for example, given the logical query $q_1$, we want to find all the entities $V_?$ such that there exists certain protein $V$ that either associate with *Alzheimer's Disease* or *Mad Cow Disease*.

CQA is challenging from the following two aspects. First, real-world KGs are always incomplete. To overcome this incompleteness issue of KGs, the task of CQA proposes to evaluate the knowledge inference capability

---

[*] Equal contribution.

| Complex Queries | Interpretations |
|---|---|
| $q_1 = V_? . \exists V : Interact(V_?, V)$ $\land (Assoc(V, Alzheimer) \lor Assoc(V, MadCow))$ | Find the substances that interact with the proteins associated with Alzheimer's or Mad cow disease. |
| $q_2 = V_? . \exists V : LocatedIn(V, America)$ $\land \neg Hold(V, WorldCup) \land IsPresident(V_?, V)$ | Listing the presidents of American countries that so far, they have not held the World Cup. |
| $q_3 = V_? : HasProfession(V_?, Fictionist)$ $\land \neg IsResident(V_?, Europe) \land Win(V_?, HugoAward)$ | Find the non-European fictionist that are the Hugo Award winners. |

Figure 1: Three complex query examples and corresponding interpretations expressed in natural language.

of a query-answering system, so that the system can still effectively answer the queries even if some of the required information needs to be implicitly inferred from the KG. The method of sub-graph matching is sensitive to the missing edges from the KG and thus unable to conduct such implicit knowledge inference (Hamilton et al., 2018; Ren et al., 2020). Second, the complexity of conducting brute-force matching is exponential to the number of variables in the complex queries. Because of these two reasons, matching algorithms cannot be directly applied to the problem of CQA (Ren et al., 2020).

Query Encoding (QE) is proposed as a fast and robust solution for CQA (Hamilton et al., 2018). Most QE systems first parse the complex query into a computational graph. The computational graph describes how to find the answers to the query by using projection and set operations. For an example shown in Figure 2 (B), suppose we want to find the substances that interact with the proteins associated with Alzheimer's or Mad Cow disease as in Figure 2 (A). In this case, we need to first find the proteins that are associated with Mad Cow disease and Alzheimer's disease respectively, then use a union operation to compute the union set of them, and finally, find what substances can interact with any of the proteins in this set.

Existing QE methods first use different neural networks to parameterize operators like *projection* and *union*, and then recursively execute them to encode a query into a query embedding (Hamilton et al., 2018). For example, they first use the embedding of *Alzheimer's Disease* and *Mad Cow Disease* and relational projection operators to compute the embeddings of two sets of proteins that are associated with them respectively. Then a *union* operator is used to compute the embedding of their union set. After this, another relation projection is used to compute the embedding of the substances that can interact with these proteins. After the encoding process is finished, they use the embedding of the answer set as the query embedding to retrieve the entities from the embedding space according to similarity scores between embeddings. In the learning process, the parameters of entity embeddings and the neural operators are jointly optimized.

Various QE methods are proposed following this paradigm, and their main focus is on using better embedding structures to encode the set of answers (Sun et al., 2020; Liu et al., 2021). For example, Ren et al. (2020); Zhang et al. (2021) propose to use geometric structures like rectangles and cones in the hyperspace to encode the entities. Bai et al. (2022a) propose to use multiple vectors to encode the query to address the diversity of the answer entities. Meanwhile, probabilistic distributions can also be used for query encoding (Choudhary et al., 2021a;b), like Beta Embedding (Ren & Leskovec, 2020) and Gamma Embedding (Yang et al., 2022).

However, the procedures of first parameterizing and then executing the operators in the graph might be potentially over-complicated, because they can be structurally simplified by using a single neural network to encode the whole computational graph. The computational graph, on the other hand, can be regarded as a special type of semantic graph telling the meaning of how to execute (Yin & Neubig, 2018; Ren et al., 2020; Ren & Leskovec, 2020). Moreover, the sequence encoders, like LSTM (Hochreiter & Schmidhuber, 1997) and transformer (Vaswani et al., 2017), achieve high performance on tasks involving semantic graph encoding, such as Graph-to-Text generation (Konstas et al., 2017; Ribeiro et al., 2021).

Inspired by this, we propose sequential query encoding (SQE) for complex query encoding. In SQE, instead of parameterizing the operators and executing the computational graph, we use a search-based algorithm to linearize the graph into a sequence of tokens. After this, SQE uses a sequence encoder, like LSTM (Hochreiter & Schmidhuber, 1997) and Transformer (Vaswani et al., 2017), to encode this sequence of tokens. Its output

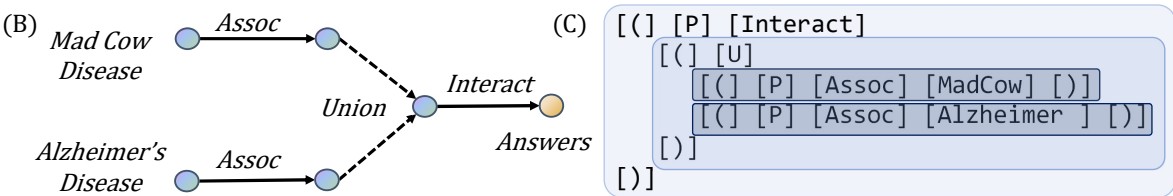

Figure 2: (A) The example complex query with the meaning of *Finding the substances that interact with the proteins associated with Alzheimer's or Mad cow disease.* (B) The computational graph of the complex query; (C) The linearized computational graph with proper indentations as a sequence of tokens, and the brake lines and indents are for better display. All the terms enclosed by square brackets `[.]` are tokens. The token of `[(]` and `[)]` are used to indicate the original graph structure. The token of `[P]` and `[U]` are tokens representing the *projection* and *union* operators. `[Interact]` and `[Assoc]` are the tokens representing relations. `[MadCow]` and `[Alzheimer]` are tokens representing the entities. All the tokens of brackets, operations, relations, and entities are treated in the same way by a sequence encoder.

sequence embedding is used as the query embedding. Similarly, SQE computes the similarity scores between the query embedding and entity embeddings to retrieve answers from the embedding space.

Despite its simplicity, SQE demonstrates better faithfulness and inference capability than state-of-the-art neural query encoders over FB15k (Bollacker et al., 2008; Bordes et al., 2013), FB15k-237 (Toutanova & Chen, 2015), and NELL (Carlson et al., 2010), under an extended benchmark, which includes twenty-nine types of logical queries (Wang et al., 2021). Moreover, we further evaluate the compositional generalization (Fodor & Lepore, 2002) of the SQE to see whether it can also generalize to the out-of-distribution query types that are unobserved during the training process. Again, the SQE method with an LSTM backbone demonstrates comparable inference capability to state-of-the-art neural query encoders on the out-of-distribution queries. [1] The main contributions of this paper can be summarized as follows:

- We propose sequential query encoding (SQE), which is the first method that uses sequence encoders to encode linearized computational graphs for answering first-order logic queries on KG.

- We conduct extensive experiments to demonstrate that SQE is the current state-of-the-art neural query encoding method for encoding in-distribution queries. Meanwhile, SQE demonstrates comparable inference capability on the out-of-distribution queries.

- Further analysis shows that even using the same neural structure, compared with executing following the computational graph, sequential encoding demonstrates better performance in encoding in-distribution queries on both faithfulness and knowledge inference capability.

## 2 Problem Definition

### 2.1 Logical Query

CQA is conducted on a knowledge graph $\mathcal{G} = (\mathcal{V}, \mathcal{R})$. The $\mathcal{V}$ is the set of vertices $v$, and the $\mathcal{R}$ is the set of relation $r$. To describe the relations in logical expressions, the relations are defined in functional forms. Each relation $r$ is defined as a function, and it has two arguments, which represent two entities $v$ and $v'$. The value of function $r(v, v') = 1$ if and only if there is a relation between the entities $v$ and $v'$.

The queries are defined in the first-order logical (FOL) forms. In a first-order logical expression, there are logical operations such as existential quantifiers $\exists$, conjunctions $\wedge$, disjunctions $\vee$, and negations $\neg$. In such

---

[1]Code available: https://github.com/HKUST-KnowComp/SQE

a logical query, there are anchor entities $V_a \in \mathcal{V}$, existential quantified variables $V_1, V_2, ...V_k \in \mathcal{V}$, and a target variable $V_? \in \mathcal{V}$. The knowledge graph query is written to find the answer entities $V_? \in \mathcal{V}$, such that there exist $V_1, V_2, ...V_k \in \mathcal{V}$ satisfying the logical expression in the query. For each query, it can be converted to a disjunctive normal form, where the query is expressed as a disjunction of several conjunctive expressions:

$$q[V_?] = V_?.\exists V_1, ..., V_k : c_1 \vee c_2 \vee ... \vee c_n, \tag{1}$$

$$c_i = e_{i1} \wedge e_{i2} \wedge ... \wedge e_{im}. \tag{2}$$

Each $c_i$ represents a conjunctive expression of literals $e_{ij}$, and each $e_{ij}$ is an atomic or the negation of an atomic expression in any of the following forms: $e_{ij} = r(v_a, V)$, $e_{ij} = \neg r(v_a, V)$, $e_{ij} = r(V, V')$, or $e_{ij} = \neg r(V, V')$. Here $v_a \in V_a$ is one of the anchor entities, and $V, V' \in \{V_1, V_2, ..., V_k, V_?\}$ are distinct variables satisfying $V \neq V'$. When a query is an existential positive first-order (EPFO) query, there are only conjunctions $\wedge$ and disjunctions $\vee$ in the expression (no negations $\neg$). When the query is a conjunctive query, there are only conjunctions $\wedge$ in the expressions (no disjunctions $\vee$ and negations $\neg$).

## 2.2 Computational Graph

As shown in Figure 2 (B), there is a corresponding computational graph for each FOL logical query. The computational graph is defined in a directional acyclic graph (DAG) structure. The nodes and edges in the graph represent the intermediate states and operations respectively. The operations are used to encode the sub-queries following the computational graph recursively, they implicitly model the set operations of the intermediate query results. The set operations are as follows:

- *Relational Projection*: Given a set of entities $A$ and a relation $r \in R$, this operation returns all entities holding relation $r$ with at least one entity $e \in A$. Namely, $P_r(A) = \{v \in \mathcal{V} | \exists v' \in A, r(v', v) = 1\}$;

- *Intersection*: Given sets of entities $A_1, ...A_n \subset \mathcal{V}$, this operation computes their intersection $\cap_{i=1}^{n} A_i$;

- *Union*: Given several sets of entities $A_1, ...A_n \subset \mathcal{V}$, this operation calculates their union $\cup_{i=1}^{n} A_i$;

- *Complement/Negation*: Given a set of entities $A$, it calculates the absolute complement $\mathcal{V} - A$.

# 3 Sequential Query Encoding

In this paper, we propose an alternative way to encode the complex query by first linearizing a computational graph into a sequence of tokens, and then using a sequence encoder to compute its sequence embedding as the corresponding query embedding. For example in Figure 2, we equivalently convert a computational graph (B) into a sequence of tokens shown in (C) by using the Algorithm 1. Then SQE uses sequence encoders, such as LSTM and transformer, to encode the token sequence in Figure (C). Finally, SQE uses the output sequence embedding as query embedding to retrieve answers from the entity embedding space.

## 3.1 Linearizing Computational Graph

A directed acyclic computational graph is first linearized to a sequence of tokens. Our linearizing algorithm as shown in Algorithm 1, starts from its target node $T$. First, we try to find the last operation in this graph for $T$. It could be either a relational *projection*, *intersection*, *union*, or *negation*. Then the first two tokens of the answer are determined as [(][P], [(][I], [(][U], and [(][N] correspondingly, and the last token is determined as [)]. If the operation type is *projection*, we will additionally add the third token indicating the relation type like [Interact] in Figure 2 (C). After this, we will find the nodes in the DAG that have an outward edge pointing to the target node $T$, and these nodes are called previous nodes. If the operation is *projection* or *negation*, there is always only one such node. If the operation is *intersection* and *union*, there might be two or more such nodes. Regardless of the operation type, the linearizing algorithm is recursively called on the previous nodes until the base case is reached. The base case of this recursion is the previous node is a given anchor entity, such as Mad Cow disease in Figure 2 (C). Then the algorithm will use a unique token to represent this entity like [MadCow]. During the recursions, the output tokens from the previous nodes are put between the square bracket tokens [(] and [)] determined by the target node.

---

**Algorithm 1** Linearization of the computational graph of a query into a sequence of tokens.

---

**Require:** $G$ is the computational graph of a certain query.
**Require:** $Tokenize()$ is the function that converts relations and entities to their token ids
    **function** LINEARIZE($T$)
        $T$ is the target node of the computational graph.
        **if** $T.operation = projection$ **then**
            `Rel` $\leftarrow Tokenize(T.relation)$
            `SubQueryTokens` $\leftarrow$ LINEARIZE($T.prev$)
            **return** `[(][P] + Rel + SubQueryTokens + [)]`
        **else if** $T.operation = intersection$ or $T.operation = union$ **then**
            `QueryTokens` $\leftarrow$ `[(]`
            **if** $T.operation = intersection$ **then**
                `QueryTokens` $\leftarrow$ `QueryTokens + [I]`
            **else**
                `QueryTokens` $\leftarrow$ `QueryTokens + [U]`
            **end if**
            **for** $prev \in T.prevs$ **do**
                `SubQueryTokens` $\leftarrow$ LINEARIZE($prev$)
                `QueryTokens` $\leftarrow$ `QueryTokens + SubQueryTokens`
            **end for**
            **return** `QueryTokens + [)]`
        **else if** $T.operation = negation$ **then**
            $SubQueryTokens \leftarrow$ LINEARIZE($T.prev$)
            **return** `[(][N] + SubQueryTokens + [)]`
        **else if** $T.operation = e$ **then return** $Tokenize(T.entity)$
        **end if**
    **end function**

---

## 3.2 Encoding Linearized Computational Graph

After the computational graph is linearized to a sequence of tokens, SQE uses a sequence encoder to encode them. All the tokens in the sequence, including the brackets `[(][)]`, operations `[P][I][N][U]`, relations `[Assoc][Interact]`, and entities `[Alzheimer]` are assigned with unique and unified ids respectively. Then a unified embedding table is created to hold the token embeddings corresponding to all these tokens.

As shown in Figure 3, the input tokens are first converted to a sequence of embedding vectors, and then these embeddings are used as input to sequence encoders. The sequence encoder will compute the contextualized representation for each token, and we use the embedding of the first token as the sequence representation. This sequence representation is used as the query embedding to retrieve answers from the entity embedding space. Suppose the entity embedding of the entity $v$ is $e_v$, and the sequence embedding of query $q$ is $e_q$. We use the inner product between $e_v$ and $e_q$ as the measurement of the similarity between query $q$ and entity $v$. To train the SQE model, we compute the normalized probability of the entity $v$ being the correct answer of query $q$ by using the `softmax` function on all similarity scores,

$$p(q, v) = \frac{e^{<e_q, e_v>}}{\sum_{v' \in V} e^{<e_q, e_{v'}>}}. \tag{3}$$

Then we construct a cross-entropy loss to maximize the log probabilities of all correct query-answer pairs:

$$L = -\frac{1}{N} \sum_i \log p(q^{(i)}, v^{(i)}). \tag{4}$$

Each $(q^{(i)}, v^{(i)})$ denotes one of the positive query-answer pairs, and there are $N$ pairs in the training set.

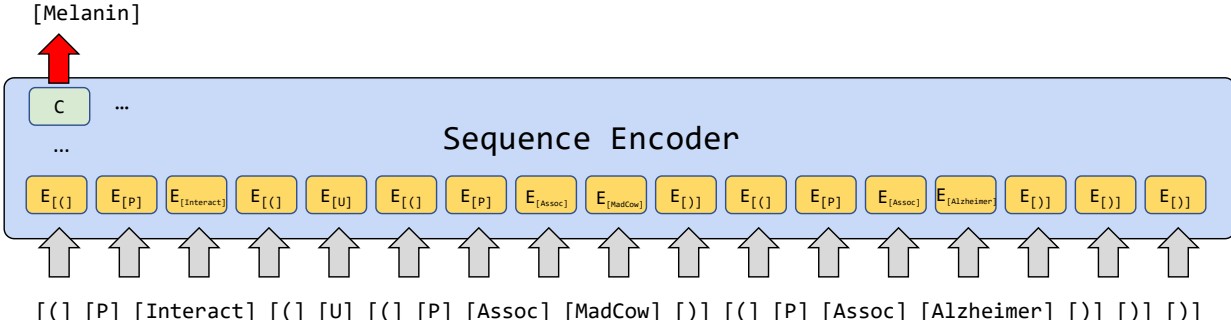

Figure 3: The illustration of using a sequence encoder to encode the linearized computational graph. The sequence encoder uses the representation of the first output token as the sequence representation.

## 4 Experiment

In this section, we first explain the evaluation settings and metrics for query encoding. Then we discuss the knowledge graphs and the benchmark datasets for evaluation. Then we briefly present the neural query encoding baseline methods that we directly compared with. After this, we disclose the implementation details for SQE. Finally, we discuss the experiment results and conduct further analysis on SQE.

### 4.1 Evaluations

In our experiment, following previous work, we also use the following three knowledge graphs: FB15k (Bollacker et al., 2008; Bordes et al., 2013), FB15k-237 (Toutanova & Chen, 2015), and NELL (Carlson et al., 2010). As shown in Table 8, the edges in each knowledge graph are separated into training, validation, and testing with a ratio of 8:1:1 respectively. Training graph $\mathcal{G}_{train}$, validation graph $\mathcal{G}_{val}$, and test graph $\mathcal{G}_{test}$ are constructed by training edges, training+validation edges, and training+validation+testing edges respectively following previous setting (Ren et al., 2020). All QE models have evaluated the following three aspects: faithfulness, knowledge inference capability, and compositional generalization.

#### 4.1.1 Faithfulness and Knowledge Inference Capability

The most important aspect of the query encoding model is its capability of knowledge `inference`. To robustly deal with the incompleteness of knowledge graphs, the query encoding model is required to answer queries with answers that need to be implicitly inferred from existing facts in the KG. On the other hand, the capability of `entailment` is also measured to evaluate whether a QE model can faithfully answer the queries that are explicitly shown on the training graph (Sun et al., 2020).

To precisely describe the metrics, we use the $q$ to represent a testing query and $\mathcal{G}_{val}$, $\mathcal{G}_{test}$ to represent the validation and the testing knowledge graph. Here we use $[q]_{val}$ and $[q]_{test}$ to represent the answers of query $q$ on the validation graph $\mathcal{G}_{val}$ and testing graph $\mathcal{G}_{test}$ respectively. Equation 5 and Equation 6 describe how to compute the `Inference` and `Entailment` metrics respectively. When the evaluation metric is Hit@K, the $m(r)$ is defined as $m(r) = \mathbf{1}[r \leq K]$. In other words, $m(r) = 1$ if $r \leq K$, otherwise $m(r) = 0$. Meanwhile, if the evaluation metric is mean reciprocal ranking (MRR), then the $m(r)$ is defined as $m(r) = \frac{1}{r}$.

$$\texttt{Inference}(q) = \frac{\sum_{v \in [q]_{test}/[q]_{val}} m(\texttt{rank}(v))}{|[q]_{test}/[q]_{val}|}. \tag{5}$$

$$\texttt{Entailment}(q) = \frac{\sum_{v \in [q]_{train}} m(\texttt{rank}(v))}{|[q]_{train}|}. \tag{6}$$

During the training process, the testing graph $\mathcal{G}_{test}$ is unobserved. In the hyper-parameters selection process, we are still using the metrics in Equation 5 but replacing graphs $\mathcal{G}_{test}/\mathcal{G}_{val}$ by $\mathcal{G}_{val}/\mathcal{G}_{train}$ respectively.

Table 1: The comparison between different benchmark datasets on the number of query types. The in-distribution query types refer to the types that the QE model is trained on during the training process. The out-of-distribution query types refer to the types that are unobserved in the training process, but are required to be tested in the evaluation process.

| Datasets | In-distribution Types | Out-of-distribution Types | Total |
|---|---|---|---|
| Q2B (Ren et al., 2020) | 5 | 4 | 9 |
| BetaE (Ren & Leskovec, 2020) | 10 | 4 | 14 |
| SMORE (Ren et al., 2022) | 10 | 4 | 14 |
| Our Benchmark | **29** | **29** | **58** |

### 4.1.2 Compositional Generalization

In addition to these two aspects, the compositional generalizability of the query encoding models should also be systematically studied. Compositionality is the idea that the meaning of a complex expression can be constructed from its less complex sub-expressions (Fodor & Lepore, 2002). The compositional generalizability describes the capability of a system that, when it is given some primitive examples and their simple combinations, the system can deal with the examples with unseen combinations. Such ability is commonly evaluated on the problem of language or visual reasoning (Johnson et al., 2017; Finegan-Dollak et al., 2018; Loula et al., 2018; Lake & Baroni, 2018; Keysers et al., 2020). In this paper, we extend the evaluation of compositional generalizability toward the problem of complex query answering.

Compositional generalizability is critical to query encoding. As the total number of query types grows exponentially with the number of variables inside a complex query, it is infeasible to enumerate all query types for training. So the QE models are expected to have compositional generalizability to enhance their performance on the unseen/out-of-distribution query types. For example, suppose the query when the QE methods are trained on the query types of `(p,(e))`, such as *What substance can interact with Prion Protein?* and `(u,(p,(e)), (p,(e)))`, like *What protein are associated with Alzheimer's or Mad Cow disease?*, we expect it can also perform well on `(p,(u,(p,(e)), (p,(e))))`, like *What are the substances that interact with the proteins associated with Alzheimer's or Mad cow disease*, which is composed of the previous two.

### 4.2 Benchmarks

Most existing QE models use the benchmark from Ren & Leskovec (2020) to evaluate their performance. However, this benchmark has two major drawbacks. First, its number of query types is limited. In total, it includes fourteen types of queries. Because of this, it is insufficient to describe the complex structures of general logical queries. Second, it cannot be used for evaluating compositional generalizability because all the query types that are evaluated involve the operators that are not observed during the training process. Recently, SMORE (Ren et al., 2022) scales up the size of the knowledge graph and the number of samples but keeps the same query types, leaving these two problems unsolved. Meanwhile, Wang et al. (2021) discusses the logic forms of different query types and scales up the query types. However, the experiments and discussions on compositional generalization are still limited to five in-distribution query types and two out-of-distribution query types. Because of these reasons, we choose to construct our own benchmark dataset.

To effectively evaluate the compositional generalizability, we use the queries with two anchors with a maximum depth of three as the training queries (Wang et al., 2021). Meanwhile, we additionally sample the same number of query types with three anchor entities and a maximum depth of three as unseen query types for the evaluation of compositional generalizability. As a result, we obtain twenty-nine types of in-distribution training queries and additional twenty-nine types of unseen/out-of-distribution evaluation queries. The detailed structures of query types are listed in Table 12. Meanwhile, we use the conjunctive query types from these queries to evaluate the query encoding model that does not intrinsically support *negation* and *union* operators, and their details are shown in Table 9. Here, we sample the knowledge graph queries according to the query types by using the Algorithm 2. The training queries are sampled from the training graph. The testing queries are sampled from the testing graph, while their training, validation, and testing answers are

Table 2: Number of queries used for each query type.

| Knowledge Graph | Training | | Validation | Testing |
| --- | --- | --- | --- | --- |
| | (p,(e)) | Other Types | All Types | All Types |
| FB15k | 273,710 | 821,130 | 8,000 | 8,000 |
| FB15k-237 | 149,689 | 449,067 | 5,000 | 5,000 |
| NELL-995 | 107,982 | 323,946 | 4,000 | 4,000 |

searched from the training, validation, and testing graphs respectively. Unlike previous benchmarks (Ren et al., 2020; Ren & Leskovec, 2020; Wang et al., 2021), we do not filter out queries with large answer sets, because the motivation for removing such queries in previous work is not well justified, and the removal potentially creates a distributional bias that the sampled queries could be less likely to include nodes that have higher degrees. Finally, the statistics of the queries sampled are shown in Table 1 and Table 2. Compared with previous benchmarks, we scaled up the number of query types from fourteen to fifty-eight types, while keeping the same amount of queries on each of the query types.

### 4.3 Baseline Models

We briefly introduce the baseline query encoding models that use various neural networks to parameterize the operators in the computational graph to recursively encode the query into various embedding structures.

- GQE (Hamilton et al., 2018) uses vectors to encode complex queries.

- Q2B (Ren et al., 2020) uses hyper-rectangles to encode complex queries.

- HYPE (Choudhary et al., 2021b) encodes the queries in a hyperbolic space.

- BetaE (Ren & Leskovec, 2020) uses Beta distributions to encode queries.

- ConE (Zhang et al., 2021) uses Cone Embeddings to handle negations.

- Q2P (Bai et al., 2022a) uses multiple vectors to encode the queries.

- Neural MLP (Mixer) (Amayuelas et al., 2022) use MLP and MLP-Mixer as the operators.

- FuzzQE (Chen et al., 2022) use fuzzy logic to represent logical operators.

The performance of GQE, Q2B, and Hype are shown in Table 3. Because they do not support *negation* and *union* operators during the query encoding process, we train and evaluate them separately on the conjunctive queries, and details are shown in Table 9. For the rest of the QE models, we evaluate them by using the queries shown in Table 12, and their results are shown in Table 4. All the baseline query encoding structures are implemented using the same latent space size of four hundred. There are also neural-symbolic query encoding methods proposed (Sun et al., 2020; Xu et al., 2022; Zhu et al., 2022). In this line of research, their query encoders refer back to the training knowledge graph to obtain symbolic information from the graph. Because of this, the query encoder is not purely learned from the training queries. As their contribution is orthogonal to our discussion on pure neural query encoders, we do not conduct direct comparisons.

### 4.4 Implementation Details for SQE

Sequential query encoding (SQE) is implemented with the backbones of established sequence encoding models. The previous dominant method for sequence encoding is recurrent neural networks, such as GRU, and LSTM (Hochreiter & Schmidhuber, 1997; Chung et al., 2014). The recurrent models are all used in a bi-directional way, and they are stacked for three layers. Meanwhile, SQE also uses the encoder part of the Transformer (Vaswani et al., 2017) as its sequence encoding backbone. We follow the conventions in

Table 3: The MRR results on answering conjunctive queries. The `Entailment` and `Inference` metrics are the higher the better. The best and second-best performances are marked in Bold and underlined.

| Datasets | Models | In-distribution Queries | | Out-of-distribution Queries | | Average | |
| --- | --- | --- | --- | --- | --- | --- | --- |
| | | Entailment | Inference | Entailment | Inference | Entailment | Inference |
| FB15K | GQE | 21.57 | 17.31 | 15.58 | 15.52 | 18.58 | 16.42 |
| | Q2B | 24.40 | 17.08 | 19.59 | 16.63 | 22.00 | 16.86 |
| | HYPE | 28.70 | 22.50 | 27.76 | 26.08 | 28.23 | 24.29 |
| | Q2P | 30.18 | 25.97 | 23.65 | 23.08 | 26.92 | 24.53 |
| | SQE + CNN | 39.61 | 29.36 | 36.33 | 29.65 | 38.98 | 29.42 |
| | SQE + GRU | 46.95 | 29.77 | 38.68 | **31.83** | 45.36 | 30.17 |
| | SQE + LSTM | 47.80 | 29.65 | **42.93** | 30.74 | **45.37** | **30.20** |
| | SQE + Transformer | **55.08** | **36.21** | 14.12 | 12.64 | 34.60 | 24.43 |
| FB15K-237 | GQE | 23.88 | 9.87 | 15.77 | 10.04 | 19.83 | 9.96 |
| | Q2B | 25.04 | 8.80 | 19.25 | 10.91 | 22.15 | 9.86 |
| | HYPE | 29.40 | 11.97 | 28.19 | 15.81 | 28.80 | 13.89 |
| | Q2P | 38.05 | 12.59 | 25.27 | 14.65 | 31.66 | 13.62 |
| | SQE + CNN | 51.21 | 13.14 | 45.51 | 17.30 | 50.10 | 13.95 |
| | SQE + GRU | 57.11 | 14.09 | 47.23 | 18.54 | 55.19 | 14.96 |
| | SQE + LSTM | 62.51 | 14.53 | **50.75** | **19.34** | **56.63** | **16.94** |
| | SQE + Transformer | **67.13** | **15.48** | 16.36 | 9.64 | 41.75 | 12.56 |
| NELL | GQE | 51.65 | 9.11 | 37.87 | 10.43 | 44.76 | 9.77 |
| | Q2B | 55.29 | 8.37 | 47.34 | 11.89 | 51.32 | 10.13 |
| | HYPE | 53.87 | 12.15 | 50.53 | 16.09 | 52.20 | 14.12 |
| | Q2P | 71.20 | 11.75 | 19.79 | 8.67 | 45.50 | 10.21 |
| | SQE + CNN | 83.58 | 12.32 | 77.58 | 16.60 | 82.39 | 13.16 |
| | SQE + GRU | 86.83 | 12.57 | 80.37 | 17.85 | 85.55 | 13.62 |
| | SQE + LSTM | 86.99 | 12.92 | **85.24** | **18.08** | **86.12** | **15.50** |
| | SQE + Transformer | **90.19** | **14.25** | 34.76 | 11.51 | 62.48 | 12.88 |

measuring compositional generalization capability, and use the same number of layers of the Transformer encoder structures (Kim & Linzen, 2020; Wu et al., 2023). Each of them is implemented with sixteen attention heads. The hidden sizes of both recurrent models and transformer are set to be four hundred to fairly compared with the baselines. Moreover, Table 11 shows the number of parameters of the baselines and the SQE models. The three-layer SQE models have comparable or fewer parameters than the previous neural QE methods. All the SQE and previous QE models are trained with the same batch size of 1024. All the experiments are conducted on the Nvidia GeForce RTX 3090 graphics cards.

## 4.5 Experiment Results

**Performance on In-distribution Query Types** Table 3 and 4 show the performance of the in-distribution query whose query types are used for training the QE models during the training process. The SQE models with LSTM and Transformer backbones constantly outperform previous QE methods on the evaluation of `Entailment` and `Inference` on both Conjunctive Queries and FOL queries. This means that, when dealing with the query types that have been used for training, the sequential encoding models are better at faithfully encoding the information in the knowledge graph. Meanwhile, they have better knowledge inference capability to answer queries that require implicit knowledge inference from the KG.

**Performance on Out-of-distribution Query Types** The out-of-distribution queries are used for measuring the compositional generalizability of QE models on both `Entailment` and `Inference`. As shown in Table 3, the SQE model is able to consistently outperform the models that are specifically used for encoding conjunctive queries, despite it has never seen their query types during training. However, as shown in

Table 4: The MRR results on answering FOL queries. Note that the `Entailment` and `Inference` metrics are the higher the better. The best and second-best performances are marked in Bold and underlined.

| Datasets | Models | In-distribution Queries | | Out-of-distribution Queries | | Average | |
|---|---|---|---|---|---|---|---|
| | | Entailment | Inference | Entailment | Inference | Entailment | Inference |
| FB15K | ConE | 30.14 | 20.58 | 23.01 | 16.04 | 26.58 | 18.31 |
| | BetaE | 34.91 | 19.01 | 25.84 | 14.84 | 30.38 | 16.93 |
| | Q2P | 43.00 | 23.34 | 27.47 | 15.51 | 35.24 | **19.43** |
| | Neural MLP | 44.18 | 21.62 | **31.37** | 15.78 | 37.78 | 18.70 |
| | + MLP Mixer | 38.17 | 20.45 | 27.81 | 15.11 | 32.99 | 17.78 |
| | FuzzQE | 34.99 | 21.14 | 27.03 | **16.36** | 31.05 | 18.78 |
| | SQE + CNN | 44.74 | 22.49 | 24.61 | 13.36 | 34.68 | 17.93 |
| | SQE + GRU | 48.64 | 22.67 | 29.36 | 15.11 | 39.00 | 18.89 |
| | SQE + LSTM | **49.17** | 23.17 | 29.32 | 14.90 | **39.25** | 19.04 |
| | SQE + Transformer | 49.13 | **25.83** | 13.39 | 8.00 | 31.26 | 16.92 |
| FB15K-237 | ConE | 36.69 | 9.75 | 28.13 | **8.82** | 32.41 | 9.29 |
| | BetaE | 32.48 | 8.30 | 22.96 | 7.29 | 27.72 | 7.80 |
| | Q2P | 52.33 | 10.17 | 32.70 | 8.62 | 42.52 | 9.40 |
| | Neural MLP | 51.09 | 10.03 | **36.85** | 8.75 | 43.97 | 9.39 |
| | + MLP Mixer | 45.19 | 10.07 | 33.03 | 8.66 | 39.11 | 9.37 |
| | FuzzQE | 45.60 | 9.07 | 34.69 | 8.32 | 40.18 | 8.70 |
| | SQE + CNN | 52.09 | 10.14 | 28.21 | 7.65 | 40.15 | 8.90 |
| | SQE + GRU | 55.46 | 10.59 | 32.25 | 8.34 | 43.86 | 9.47 |
| | SQE + LSTM | 56.02 | 10.62 | 33.41 | 8.62 | **44.72** | **9.62** |
| | SQE + Transformer | **59.15** | **11.30** | 15.06 | 4.98 | 37.11 | 8.14 |
| NELL | ConE | 58.36 | 8.55 | 45.07 | 7.92 | 51.72 | 8.24 |
| | BetaE | 48.13 | 7.06 | 34.63 | 6.65 | 41.38 | 6.86 |
| | Q2P | 79.79 | 10.29 | 56.18 | 8.45 | 67.99 | 9.37 |
| | Neural MLP | 80.42 | 9.98 | 63.12 | 8.20 | 71.77 | 9.09 |
| | + MLP Mixer | 78.08 | 10.05 | 58.67 | 8.26 | 68.38 | 9.16 |
| | FuzzQE | 77.81 | 8.63 | **63.58** | 7.60 | 70.71 | 8.12 |
| | SQE + CNN | 82.10 | 9.99 | 51.80 | 7.30 | 66.95 | 8.65 |
| | SQE + GRU | 85.36 | 10.30 | 57.42 | 8.21 | 71.39 | 9.26 |
| | SQE + LSTM | 85.42 | 10.37 | 60.06 | **8.59** | **72.74** | **9.48** |
| | SQE + Transformer | **85.52** | **10.95** | 22.24 | 4.97 | 53.88 | 7.96 |

Table 4, for FOL queries, when SQE is used for the query types that have not been used for training, it performs worse on `Entailment` than previous models. This implies that SQE models are less compositional generalizable on faithfulness on FOL queries. However, SQE performs comparably to previous QE methods in the `Inference` metric on out-of-distribution query types. This indicates SQE is comparably compositional and generalizable to the previous QE model on the knowledge inference capability on FOL queries. The performance differences in conjunctive queries and FOL queries can be explained by the structural differences. The conjunctive queries only contain `intersection` and `projection`, but FOL queries contain `intersection`, `union`, `negation`, and `projection`. Therefore conjunctive queries are structurally simpler, and their structural information is easier to be captured by sequence encoders, which leads to higher compositional generalization.

## 5 Discussion

There are two major differences between the previous QE methods and SQE: (1) The previous query encoding methods encode the query recursively following the computational graph, but the sequence encoder directly uses special input tokens of `[(]` and `[)]` to indicate and represent the graph structure as model input; (2)

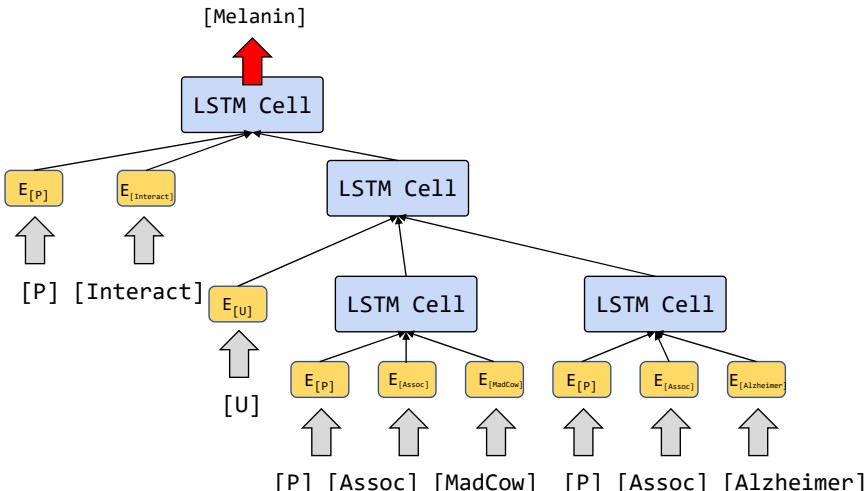

Figure 4: A Tree-LSTM encoder (Tai et al., 2015) is used to encode the computational DAG. All the operations, relations, and entities are converted to corresponding token embeddings.

The previous query encoding contains the inductive bias stemming from the understanding of logical and set operations in parameterizations and encoding structures, while the sequence encoder also regards logical operations as some special tokens, whose meaning and functionality are purely learned from the empirical data. To further investigate the effects of these factors on the performance differences, we further propose to use another semantic graph encoding model to control the variables. Tree-LSTM (Tai et al., 2015) is another model that is widely used in NLP tasks to encode semantic and syntactic parsing graphs. As shown in Figure 4, Tree-LSTM can also encode the computational graph in CQA recursively following the computational graph, which is the same as previous QE methods. On the other hand, Tree-LSTM treated all operations, entities, and relations as tokens. Thus, similar to SQE, Tree-LSTM does not have prior knowledge and related structural inductive bias about logic and set operations.

## 5.1 The importance of using sequential query encoding

Both the LSTM and Tree-LSTM models use the same encoding unit of LSTM cells. However, in the LSTM model, the LSTM cells are sequentially connected, whereas in the Tree-LSTM model, the LSTM cells are connected from the leaves to the root following a tree structure. As shown in Table 5, SQE+LSTM performs consistently better than the Tree-LSTM on three datasets on `Entailment` and `Inference` over queries on the in-distribution queries. This indicates that when using the same encoding structure, the sequence encoding used in SQE is better than encoding recursively following the computational graph on in-distribution queries. Meanwhile, for the out-of-distribution queries, the faithfulness of Tree-LSTM is better than SQE+LSTM, but they have comparable knowledge inference capabilities.

When using SQE+LSTM, the structural information of complex queries is encoded in the input tokens and is purely learned in the optimization process. However, for the Tree-LSTM model, the structural information, i.e., how to connect the LSTM cells, is directly given even for queries whose types are not observed during the training process. This can explain why Tree-LSTM performs better than SQE+LSTM on out-of-distribution queries. However, for in-distribution queries, the problem of query encoding is a pure sequence encoding task, and the LSTM model is already trained on how to leverage the structural information on the observed query types. Therefore, LSTM performs better than Tree-LSTM on in-distribution queries.

## 5.2 The importance of neural network designs in query encoding

To further explore the reason why Tree-LSTM is able to achieve comparable and even better performance than the previous query encoding methods, we conduct ablations on its structure by removing its memory cell vector $c_i$ in the LSTM cell. The performance of Tree-LSTM drastically drops when the memory cells

Table 5: Comparison between Tree-LSTM query encoder and SQE+LSTM encoder. The Memory cell removed version LSTM cell suffers vanishing gradients.

| Datasets | Models | In-distribution Queries | | Out-of-distribution Queries | | Average | |
| | | Entailment | Inference | Entailment | Inference | Entailment | Inference |
|---|---|---|---|---|---|---|---|
| FB15K | SQE + LSTM | **49.17** | **23.17** | 29.32 | 14.90 | **39.25** | **19.04** |
| | Tree LSTM | 46.42 | 21.50 | **31.89** | **15.71** | 39.16 | 18.61 |
| | - Memory Cell | 33.78 | 17.26 | 20.25 | 11.75 | 27.02 | 14.51 |
| FB15K-237 | SQE + LSTM | **56.02** | **10.62** | 33.41 | **8.62** | 44.72 | **9.62** |
| | Tree LSTM | 53.15 | 9.63 | **36.89** | 8.61 | **45.02** | 9.12 |
| | - Memory Cell | 39.17 | 8.63 | 21.80 | 7.27 | 30.49 | 7.95 |
| NELL | SQE + LSTM | **85.42** | **10.37** | 60.06 | **8.59** | 72.74 | **9.48** |
| | Tree LSTM | 85.39 | 9.79 | **68.52** | 8.28 | **76.96** | 9.04 |
| | - Memory Cell | 67.33 | 8.03 | 28.89 | 5.58 | 48.11 | 6.81 |

Table 6: The performance comparison between BiQE and SQE+Transformer on conjunctive queries.

| Datasets | Models | In-distribution Queries | | Out-of-distribution Queries | | Average | |
| | | Entailment | Inference | Entailment | Inference | Entailment | Inference |
|---|---|---|---|---|---|---|---|
| FB15K | BiQE | 52.73 | 35.64 | **25.53** | **21.41** | **39.13** | **28.53** |
| | SQE + Transformer | **55.08** | **36.21** | 14.12 | 12.64 | 34.60 | 24.43 |
| FB15K-237 | BiQE | 64.35 | 14.86 | **39.36** | **16.21** | **51.86** | **15.54** |
| | SQE + Transformer | **67.13** | **15.48** | 16.36 | 9.64 | 41.75 | 12.56 |
| NELL | BiQE | 88.87 | **14.29** | **37.79** | 10.83 | **63.33** | 12.56 |
| | SQE + Transformer | **90.19** | 14.25 | 34.76 | **11.51** | 62.48 | **12.88** |

are removed. The memory cells in LSTM cells are mainly designed to prevent gradient vanishing during optimization. The performance drop also suggests potential optimization issues for the recursive query encoders. This is because previous QE methods are designed based on the understanding of logic and set operations. However, the design of the previous query encoding structures neglects whether the structures proposed can be effectively optimized without experiencing optimization issues. It is also possible that, although a QE has a good inductive bias for encoding logic and sets, it may not be effectively optimized due to gradient vanishing or gradient explosion. The SQE model, on the other hand, uses established sequence encoders as backbones and is less likely to suffer from technical problems in optimization.

## 5.3 The importance of query representation

Previous research on QE proposed various ways to represent complex queries. For example, Ren et al. (2020) propose to use hyper-rectangles, and Bai et al. (2022a) propose to use particle embeddings to represent the complex queries. Meanwhile, vector embeddings are generally perceived as insufficient to represent the answers to complex queries. However, in the SQE method, each query is simply represented as a vector embedding. In addition to this, Tree-LSTM also uses a single vector to achieve comparable results to previous QE methods. This indicates that, with the proper design of neural network structures and effective optimization, vector embedding is also able to achieve comparable or better performance.

## 5.4 Why SQE-Transformer bad at compositional generalization?

We notice that the compositional generalizability of the SQE+Transformer encoding is low. So we conduct further experiments and analysis on why it has such a performance drop. We compare the sequence encoder Transformer and another Transformer-based encoding method. BiQE (Kotnis et al., 2021) also proposes to use a transformer to encode complex queries. Differently, they use special positional encoding schemes

Table 7: Improving the compositional generalizability of Transformer by using relative positional encoding.

| Datasets | Models | In-distribution Queries | | Out-of-distribution Queries | | Average | |
| --- | --- | --- | --- | --- | --- | --- | --- |
| | | Entailment | Inference | Entailment | Inference | Entailment | Inference |
| FB15K | SQE + Transformer | **49.13** | **25.83** | 13.39 | 8.00 | 31.26 | 16.92 |
| | + Relative PE | 48.76 | 25.45 | **25.74** | **14.55** | **37.35** | **20.06** |
| FB15K-237 | SQE + Transformer | **59.15** | **11.30** | 15.06 | 4.98 | 37.11 | 8.14 |
| | + Relative PE | 56.97 | 11.13 | **28.75** | **7.78** | **42.96** | **9.47** |
| NELL | SQE + Transformer | 85.52 | **10.95** | 22.24 | 4.97 | 53.88 | 7.96 |
| | + Relative PE | **87.59** | 10.85 | **48.91** | **7.35** | **68.28** | **9.11** |

instead of tokens to represent the graph structure and operations. Because of its specially designed positional encoding scheme, BiQE is only applicable to conjunctive queries and is unable to deal with *union* and *negation* operators. To conduct fair comparisons, we use the same implementation of the transformer structure with the same number of parameters under the same training and evaluation setting. According to Table 6, BiQE performs better on out-of-distribution queries and worse on in-distribution queries than the SQE+Transformer model, both on `Entailment` and `Inference`. This indicates that the original positional encoding of the Transformer is the main reason for the poor compositional generalizability on complex query answering.

### 5.5 Improving compositional generalizability of Transformer for Sequential Query Encoding

Various methods are proposed to improve the compositional generalizability of sequence-to-sequence models for semantic parsing (Lake & Baroni, 2018; Hupkes et al., 2020; Jacob et al., 2022). Ontanon et al. (2022) investigated different methods to enhance the Transformer model. They demonstrated that incorporating relative position encoding (RPE), using a copy decoder, and adding intermediate representations can be beneficial. For the specific task of CQA, the Transformer model is utilized solely for the encoding process and the intermediate representations are not applicable to CQA. Therefore, our focus is on utilizing relative positional encoding to enhance the Transformer model's performance on CQA. Relative positional encoding (RPE) is introduced by Shaw et al. (2018). The RPE is a modification to the original positional encoding used in the Transformer. While the original encoding used embeddings that corresponded to the positions of input tokens, the RPE incorporates positional information by adding a learnable relative position embedding to the attention modules of the transformer layers. Experimental results demonstrate that, with relative position encoding, the SQE + Transformer model exhibits substantial improvement in out-of-distribution queries while maintaining comparable performance on in-distribution queries.

## 6 Related Work

Complex query answering is a deductive knowledge graph reasoning task, in which a model or system is required to answer the logical query on an incomplete knowledge graph. Query encoding is a fast and robust method for dealing with complex query answering. Recently, there is also new progress on query encoding that is orthogonal to this paper, which puts a focus on the neural encoders for complex queries. Xu et al. (2022) propose a neural-symbolic entangled method, ENeSy, for query encoding. Yang et al. (2022) propose to use Gamma Embeddings to encode complex logical queries. Liu et al. (2022) propose to use pre-training on the knowledge graph with kg-transformer and then conduct fine-tuning on the complex query answering.

Meanwhile, query decomposition (Arakelyan et al., 2021) is another way to deal with the problem of complex query answering. In this method, the probabilities of atomic queries are first computed by a link predictor, and then continuous optimization or beam search is used to conduct inference time optimization. Moreover, Wang et al. (2023) propose an alternative to query encoding and query decomposition, in which they conduct message passing on the one-hop atomics to conduct complex query answering. Recently a novel neural search-based method QTO (Bai et al., 2022b) is proposed. QTO demonstrates impressive performance CQA. However, its worst-case reasoning efficiency is quadratic to the size of KG. Xi et al. (2022) propose

ROMA, a framework to answer complex logical queries on multi-view knowledge graphs. Theorem proving is another deductive reasoning task on knowledge graphs. Bai et al. (2023b) propose the task of numerical CQA and the corresponding solution of number reasoning network, which can effectively deal with the numerical values in the KGs in the reasoning process. Moreover, Bai et al. (2023a) study differences between reasoning over entity-centric KG and eventuality-centric KG, and formulate the reasoning problem over knowledge graph describing event, states, and actions. Neural theorem proving (Rocktäschel & Riedel, 2017; Minervini et al., 2020; 2021) methods are proposed to deal with the incompleteness of existing knowledge graphs to conduct inference on the missing information by using embeddings.

## 7   Conclusions

In this paper, we present sequential query encoding for complex query answering. We evaluate the faithfulness and inference capability for various query encoding methods on both in-distribution and out-of-distribution queries. Experiments show that, despite its simplicity, SQE has better faithfulness and inference capability than existing neural query encoding methods on in-distribution query types. Meanwhile, SQE achieves comparable inference capability to previous QE methods on out-of-distribution query types. Further experiments address the importance of the structural design of neural network structures, meanwhile demonstrating that the existing design of positional encoding in Transformer obstructs its compositional generalization to out-of-distribution queries.

## Acknowledgement

The authors of this paper are supported by the NSFC Fund (U20B2053) from the NSFC of China, the RIF (R6020-19 and R6021-20), and the GRF (16211520 and 16205322) from RGC of Hong Kong, the MHKJFS (MHP/001/19) from ITC of Hong Kong and the National Key R&D Program of China (2019YFE0198200) with special thanks to HKMAAC and CUSBLT. We also thank the UGC Research Matching Grants (RMGS20EG01-D, RMGS20CR11, RMGS20CR12, RMGS20EG19, RMGS20EG21, RMGS23CR05, RMGS23EG08).

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

## A   Sampling Algorithm

In this section, we introduce the algorithm used for sampling the complex queries from a given knowledge graph. The detailed algorithm is described in Algorithm 2. For a given knowledge Graph $G$ and a query type $t$, we start with a random node $v$ to reversely find a query that has answer $v$ with the corresponding structure $t$. Basically, this process is conducted in a recursion process. In this recursion, we first look at the last operation in this query. If the operation is *projection*, we randomly select one of its predecessors $u$ that holds the corresponding relation to $v$ as the answer of its sub-query. Then we call the recursion on node $u$ and the sub-query type of $t$ again. Similarly, for *intersection* and *union*, we will apply recursion on their sub-queries on the same node $v$. The recursion will stop when the current node contains an anchor entity.

---

**Algorithm 2** Ground Query Type

---

**Require:** $G$ is a knowledge graph.
   **function** GROUNDTYPE$(T, v)$
      $T$ is an arbitrary node of the computation graph.
      $v$ is an arbitrary knowledge graph vertex
      **if** $T.operation = p$ **then**
         $u \leftarrow$ SAMPLE$(\{u|(u,v)\text{is an edge in } G\})$
         $RelType \leftarrow$ type of $(u, v)$ in $G$
         $ProjectionType \leftarrow p$
         $SubQuery \leftarrow$ GROUNDTYPE$(T.child, u)$
         **return** $(ProjectionType, RelType, SubQuery)$
      **else if** $T.operation = i$ **then**
         $IntersectionResult \leftarrow (i)$
         **for** $child \in T.Children$ **do**
            $SubQuery \leftarrow$ GROUNDTYPE$(T.child, v)$
            $IntersectionResult.$PUSHBACK$(child, v)$
         **end for**
         **return** $IntersectionResult$
      **else if** $T.operation = u$ **then**
         $UnionResult \leftarrow (u)$
         **for** $child \in T.Children$ **do**
            **if** $UnionResult.length > 2$ **then**
               $v \leftarrow$ SAMPLE$(G)$
            **end if**
            $SubQuery \leftarrow$ GROUNDTYPE$(T.child, v)$
            $UnionResult.$PUSHBACK$(child, v)$
         **end for**
         **return** $UnionResult$
      **else if** $T.operation = e$ **then**
         **return** $(e, T.value)$
      **end if**
   **end function**

---

Table 8: The basic information about the three knowledge graphs used for the experiments, and their standard training, validation, and testing edges separation according to Ren & Leskovec (2020).

| Dataset | Relations | Entities | Training | Validation | Testing | All Edges |
|---------|-----------|----------|----------|------------|---------|-----------|
| FB15k | 1,345 | 14,951 | 483,142 | 50,000 | 59,071 | 592,213 |
| FB15k-237 | 237 | 14,505 | 272,115 | 17,526 | 20,438 | 310,079 |
| NELL995 | 200 | 63,361 | 114,213 | 14,324 | 14,267 | 142,804 |

Table 9: In-distribution and out-of-distribution query types in conjunctive queries for Table 3.

| Conjunctive Queries | Number of Types | Query Formula | Query Depth |
|---|---|---|---|
| In-Distribution | 12 | (p,(e)) | 1 |
| | | (p,(p,(e))) | 2 |
| | | (p,(p,(p,(e)))) | 3 |
| | | (p,(i,(p,(e)),(p,(e)))) | 2 |
| | | (p,(i,(p,(e)),(p,(p,(e))))) | 3 |
| | | (p,(i,(p,(p,(e))),(p,(p,(e))))) | 3 |
| | | (i,(p,(e)),(p,(e))) | 1 |
| | | (i,(p,(e)),(p,(p,(e)))) | 2 |
| | | (i,(p,(e)),(p,(p,(p,(e))))) | 3 |
| | | (i,(p,(p,(e))),(p,(p,(e)))) | 2 |
| | | (i,(p,(p,(e))),(p,(p,(p,(e))))) | 3 |
| | | (i,(p,(p,(p,(e)))),(p,(p,(p,(e))))) | 3 |
| Out-of-Distribution | 3 | (i,(i,(p,(e)),(p,(p,(p,(e))))),(p,(p,(e)))) | 3 |
| | | (i,(i,(p,(e)),(p,(p,(e)))),(p,(p,(p,(e))))) | 3 |
| | | (i,(i,(p,(p,(e))),(p,(p,(p,(e))))),(p,(p,(e)))) | 3 |

Table 10: Comparisons with CQD methods on inference capability on conjunctive queries.

| Models | FB15k | | FB15k-237 | | NELL | |
|---|---|---|---|---|---|---|
| | In-dist. | Out-of-dist. | In-dist. | Out-of-dist. | In-dist. | Out-of-dist. |
| CQD-CO | 16.93 | 15.98 | 9.69 | 12.68 | 10.91 | 11.87 |
| CQD-Beam | 23.24 | 22.30 | 11.58 | 15.01 | 12.46 | 12.64 |
| GQE | 17.31 | 15.52 | 9.87 | 10.04 | 9.11 | 10.43 |
| Q2B | 17.08 | 16.63 | 8.80 | 10.91 | 8.37 | 11.89 |
| HypE | 22.50 | 26.08 | 11.97 | 15.81 | 12.15 | 16.09 |
| Q2P | 25.97 | 23.08 | 12.59 | 14.65 | 11.75 | 8.67 |
| SQE+CNN | 29.36 | 29.65 | 13.14 | 17.30 | 12.32 | 16.60 |
| SQE+GRU | 29.77 | **31.83** | 14.09 | 18.54 | 12.57 | 17.85 |
| SQE+LSTM | 29.65 | 30.74 | 14.53 | **19.34** | 12.92 | **18.08** |
| SQE+Transformer | **36.21** | 12.64 | **15.48** | 9.64 | **14.25** | 11.51 |

## B Comparison with complex query decomposition (CQD)

Table 10 shows the performance comparison between the complex query decomposition method (Arakelyan et al., 2021). We evaluated the performance of CQD, a search-based method that utilizes pre-trained link predictors for inference-time optimization, on our benchmark of conjunctive queries. To conduct this evaluation, we used the open-sourced code of CQD and their pre-trained link predictors and reported the results in Table (Arakelyan et al., 2021). As CQD does not support the negation operator, we evaluated its performance on our benchmark of conjunctive queries. It should be noted that CQD is only trained on 1p queries, so we used the terms "in-distribution" and "out-of-distribution" to indicate the sets of query types evaluated. Our experimental results show that while the CQD model outperformed some baseline models, it was still unable to outperform SQE methods.

Table 11: Comparison on model size between baseline models and the SQE models. The three-layer SQE models have a less or a comparable number of parameters than previous query encoding models.

| | Models | Number of Parameters (Million) |
|---|---|---|
| Baselines | GQE | 6.2 |
| | Q2B | 6.6 |
| | MLP | 7.6 |
| | BetaE | 13.7 |
| | ConE | 18.9 |
| SQE Models (three layers) | SQE + CNN | 7.1 |
| | SQE + GRU | 8.1 |
| | SQE + LSTM | 8.8 |
| | SQE + Transformer | 14.7 |

Table 12: In-distribution and out-of-distribution query types in first-order logic queries for Table 4.

| FOL Queries | Number of Types | Query Formula | Query Depth |
|---|---|---|---|
| In-Distribution | 29 | (p,(e)) | 1 |
| | | (p,(p,(e))) | 2 |
| | | (p,(p,(p,(e)))) | 3 |
| | | (p,(i,(p,(e)),(p,(e)))) | 2 |
| | | (p,(i,(p,(e)),(p,(p,(e))))) | 3 |
| | | (p,(i,(n,(p,(e))),(p,(e)))) | 2 |
| | | (p,(i,(p,(p,(e))),(p,(p,(e))))) | 3 |
| | | (p,(i,(n,(p,(e))),(p,(p,(e))))) | 3 |
| | | (p,(u,(p,(e)),(p,(e)))) | 2 |
| | | (p,(u,(p,(e)),(p,(p,(e))))) | 3 |
| | | (p,(u,(p,(p,(e))),(p,(p,(e))))) | 3 |
| | | (i,(p,(e)),(p,(e))) | 1 |
| | | (i,(p,(e)),(p,(p,(e)))) | 2 |
| | | (i,(p,(e)),(p,(p,(p,(e))))) | 3 |
| | | (i,(n,(p,(e))),(p,(e))) | 1 |
| | | (i,(n,(p,(p,(e)))),(p,(e))) | 2 |
| | | (i,(p,(p,(e))),(p,(p,(e)))) | 2 |
| | | (i,(p,(p,(e))),(p,(p,(p,(e))))) | 3 |
| | | (i,(n,(p,(e))),(p,(p,(e)))) | 2 |
| | | (i,(n,(p,(p,(e)))),(p,(p,(e)))) | 2 |
| | | (i,(p,(p,(p,(e)))),(p,(p,(p,(e))))) | 3 |
| | | (i,(n,(p,(e))),(p,(p,(p,(e))))) | 3 |
| | | (i,(n,(p,(p,(e)))),(p,(p,(p,(e))))) | 3 |
| | | (u,(p,(e)),(p,(e))) | 1 |
| | | (u,(p,(e)),(p,(p,(e)))) | 2 |
| | | (u,(p,(e)),(p,(p,(p,(e))))) | 3 |
| | | (u,(p,(p,(e))),(p,(p,(e)))) | 2 |
| | | (u,(p,(p,(e))),(p,(p,(p,(e))))) | 3 |
| | | (u,(p,(p,(p,(e)))),(p,(p,(p,(e))))) | 3 |
| Out-of-Distribution | 29 | (i,(i,(p,(e)),(p,(p,(p,(e))))),(p,(p,(e)))) | 3 |
| | | (u,(p,(e)),(p,(i,(n,(p,(e))),(p,(e))))) | 2 |
| | | (p,(u,(i,(n,(p,(e))),(p,(e))),(p,(e)))) | 2 |
| | | (i,(n,(p,(e))),(p,(u,(p,(e)),(p,(p,(e)))))) | 3 |
| | | (p,(i,(p,(e)),(u,(p,(p,(e))),(p,(p,(e)))))) | 3 |
| | | (i,(p,(p,(p,(e)))),(p,(u,(p,(e)),(p,(p,(e)))))) | 3 |
| | | (u,(i,(n,(p,(e))),(p,(p,(p,(e))))),(p,(p,(p,(e))))) | 3 |
| | | (i,(i,(p,(e)),(p,(p,(e)))),(p,(p,(p,(e))))) | 3 |
| | | (i,(n,(u,(p,(e)),(p,(e)))),(p,(p,(e)))) | 2 |
| | | (u,(i,(p,(e)),(p,(p,(e)))),(p,(p,(e)))) | 2 |
| | | (i,(p,(e)),(u,(p,(p,(p,(e)))),(p,(p,(p,(e)))))) | 3 |
| | | (p,(i,(i,(n,(p,(e))),(p,(p,(e)))),(n,(p,(e))))) | 3 |
| | | (u,(i,(p,(p,(e))),(p,(p,(p,(e))))),(p,(p,(p,(e))))) | 3 |
| | | (p,(u,(i,(p,(e)),(p,(p,(e)))),(p,(p,(e))))) | 3 |
| | | (i,(i,(p,(p,(e))),(p,(p,(p,(e))))),(p,(p,(e)))) | 3 |
| | | (i,(n,(p,(p,(e)))),(p,(i,(p,(e)),(p,(e))))) | 2 |
| | | (i,(p,(p,(e))),(u,(p,(e)),(p,(e)))) | 2 |
| | | (i,(p,(e)),(u,(p,(p,(e))),(p,(p,(p,(e)))))) | 3 |
| | | (u,(i,(n,(p,(e))),(p,(p,(p,(e))))),(p,(p,(e)))) | 3 |
| | | (i,(i,(p,(e)),(p,(p,(p,(e))))),(n,(p,(p,(e))))) | 3 |
| | | (u,(i,(p,(p,(e))),(p,(p,(e)))),(p,(p,(p,(e))))) | 3 |
| | | (i,(i,(p,(p,(e))),(p,(p,(p,(e))))),(n,(p,(p,(e))))) | 3 |
| | | (u,(i,(p,(e)),(p,(e))),(p,(p,(e)))) | 2 |
| | | (u,(p,(i,(n,(p,(e))),(p,(p,(e))))),(p,(p,(e)))) | 3 |
| | | (i,(p,(e)),(p,(i,(n,(p,(e))),(p,(p,(e)))))) | 3 |
| | | (u,(p,(p,(e))),(p,(u,(p,(p,(e))),(p,(p,(e)))))) | 3 |
| | | (i,(n,(p,(e))),(u,(p,(p,(e))),(p,(p,(e))))) | 2 |
| | | (p,(i,(n,(p,(e))),(u,(p,(e)),(p,(p,(e)))))) | 3 |
| | | (i,(n,(i,(n,(p,(e))),(p,(e)))),(p,(p,(p,(e))))) | 3 |

Table 13: Experiment results on various query depths on in-distribution queries.

| Datasets | Models | depth = 1 | | depth = 2 | | depth = 3 | |
|---|---|---|---|---|---|---|---|
| | | entailment | inference | entailment | inference | entailment | inference |
| FB15K | ConE | 33.13 | 27.69 | 26.90 | 18.09 | 31.31 | 19.31 |
| | MLP | 58.46 | 34.95 | 40.74 | 19.00 | 41.88 | 18.29 |
| | FuzzQE | 50.33 | 33.92 | 33.30 | 19.19 | 30.65 | 17.45 |
| | SQE + CNN | 60.50 | 35.16 | 42.55 | 20.41 | 41.08 | 18.95 |
| | SQE + GRU | 66.14 | 38.97 | 45.01 | 19.66 | 45.32 | 18.44 |
| | SQE + LSTM | **66.63** | 39.03 | 45.32 | 20.21 | **46.06** | 19.03 |
| | SQE + Transformer | 65.39 | **40.52** | **46.81** | **23.62** | 45.27 | **21.46** |
| FB15K-237 | ConE | 44.80 | 12.48 | 33.04 | 9.31 | 36.71 | 8.89 |
| | MLP | 74.47 | 12.09 | 47.21 | 9.58 | 46.42 | 9.51 |
| | FuzzQE | 69.86 | 11.07 | 44.57 | 8.53 | 38.15 | 8.58 |
| | SQE + CNN | 76.47 | 12.27 | 49.71 | 9.82 | 45.97 | 9.36 |
| | SQE + GRU | 80.65 | 12.75 | 51.30 | 10.08 | 50.34 | 9.93 |
| | SQE + LSTM | 80.78 | 12.60 | 52.30 | 10.19 | 50.70 | 9.95 |
| | SQE + Transformer | **82.81** | **13.61** | **57.50** | **10.74** | **52.62** | **10.60** |
| NELL | ConE | 71.31 | 10.30 | 55.52 | 8.38 | 56.32 | 7.96 |
| | MLP | 93.69 | 11.95 | 81.11 | 9.54 | 75.80 | 9.40 |
| | FuzzQE | 94.46 | 10.08 | 79.49 | 8.48 | 71.40 | 8.05 |
| | SQE + CNN | 95.84 | 11.66 | 83.46 | 9.79 | 76.80 | 9.36 |
| | SQE + GRU | 95.91 | 12.45 | 85.80 | 9.88 | **81.73** | 9.67 |
| | SQE + LSTM | 95.51 | 12.21 | 85.49 | 10.00 | 80.98 | 9.75 |
| | SQE + Transformer | **96.00** | **12.47** | **86.59** | **10.53** | 81.40 | **10.49** |

Table 14: Experiment results on various query depths on out-of-distribution queries.

| Datasets | Models | depth = 2 | | depth = 3 | |
|---|---|---|---|---|---|
| | | entailment | inference | entailment | inference |
| FB15K | ConE | 20.39 | 13.82 | 24.00 | 16.67 |
| | MLP | **28.49** | 13.32 | **32.74** | 16.52 |
| | FuzzQE | 25.40 | **14.49** | 27.65 | **16.91** |
| | SQE + CNN | 20.02 | 10.13 | 26.36 | 14.45 |
| | SQE + GRU | 25.19 | 11.15 | 30.94 | 16.44 |
| | SQE + LSTM | 24.38 | 10.73 | 31.20 | 16.31 |
| | SQE + Transformer | 9.88 | 5.63 | 14.72 | 8.78 |
| FB15K-237 | ConE | 25.98 | **7.22** | 28.95 | 9.31 |
| | MLP | **34.55** | 7.13 | **37.36** | 9.31 |
| | FuzzQE | 34.53 | 6.56 | 34.75 | 8.89 |
| | SQE + CNN | 22.80 | 5.84 | 30.27 | 8.24 |
| | SQE + GRU | 26.57 | 5.72 | 34.42 | 9.22 |
| | SQE + LSTM | 29.55 | 6.16 | 34.88 | **9.42** |
| | SQE + Transformer | 11.21 | 3.53 | 16.52 | 5.42 |
| NELL | ConE | 42.95 | 6.66 | 45.87 | 8.30 |
| | MLP | 64.12 | 6.84 | 62.74 | 8.58 |
| | FuzzQE | **65.40** | 6.55 | **62.89** | 7.93 |
| | SQE + CNN | 46.52 | 5.76 | 53.81 | 7.79 |
| | SQE + GRU | 54.89 | 6.13 | 58.38 | 8.87 |
| | SQE + LSTM | 60.61 | **6.88** | 60.31 | **9.10** |
| | SQE + Transformer | 17.60 | 3.45 | 24.01 | 5.44 |

