# OpenReview forum: "Sequential Query Encoding for Complex Query Answering on Knowledge Graphs"
_TMLR — Accepted by TMLR_

### Review · Reviewer_u3jx · 2023-03-29

**Summary Of Contributions:**

This paper proposes a novel method named sequential query encoding (SQE) to deal with Complex Query Answering (CQA) on knowledge graphs. The proposed method, SQE, converts a logical query into a sequence of tokens via a search-based algorithm and employs a sequence encoder to compute its vector representation, which is then used to retrieve answers from the embedding space based on similarity scores. SQE is simpler and more effective than existing methods that use neural networks to parameterize and execute the operators in the computational graph of the query. SQE achieves state-of-the-art performance on several benchmarks, including FB15k, FB15k-237, and NELL, on an extended benchmark that entails twenty-nine types of in-distribution queries. Moreover, it exhibits decent compositional generalization capabilities on out-of-distribution queries.


**Audience:**

Yes

**Broader Impact Concerns:**

There is no explicit concern to me.

**Claims And Evidence:**

Yes

**Requested Changes:**

1. The authors employ four sequence encoders (CNN, GRU, LSTM, Transformer) to demonstrate the MRR results in Table 4, but only two models (LSTM, Transformer) in Table 3. Why do the authors choose to disregard the other two models in Table 3? Is there a particular explanation?

2. There are some missing related works such as [1], it would be better to add more relevant discussions.

3. Minor comments on writing:

(1)	Last line in Page 3 (Section 2.1): What does “EPFO” stand for? A full name or a brief explanation in footnote is necessary.

(2)	First line in Section 3.1: ... a sequence of tokens Our linearizing ... -> ... a sequence of tokens. Our linearizing ...

(3)	In Section 5.1, the analysis is based on the results of Table 5, but there is no reference to it.

[1] Yushi Bai, Xin Lv, Juanzi Li, Lei Hou. Answering Complex Logical Queries on Knowledge Graphs via Query Computation Tree Optimization.

**Strengths And Weaknesses:**

Strength

1. The idea of linearizing computational graphs into vector representations is novel and intriguing in CQA on knowledge graphs, which requires structured logical queries.

2. This paper provides a clear and detailed description of its problem definition, algorithm, and evaluation settings.

3. The authors construct an extended benchmark dataset to include more in-distribution and out-of-distribution query types, which can be utilized for future research.

Weakness

1. A lack of clarification of technical design.

- The authors employ a three-layer architecture for the recurrent neural networks. I wonder whether this design is intended for a fair comparison with prior works. Perhaps it would be beneficial to provide clarification or conduct further analysis on the impact of varying the number of layers.

- The authors use three layers of the transformer encoder structures as well. The intuition is somewhat unclear and perplexing. Why do the authors choose to use the same number of layers for both the recurrent models and transformers, given that their structures are quite different?

2. Confusion regarding the analysis of experimental results.

- The authors' performance analysis on out-of-distribution query types in Section 4.5 seems incomplete as they only focus on Table 4 and ignore Table 3. As a result, the statement claiming that SQE "performs worse on Entailment than previous models" and "performs comparably to previous QE methods in the Inference metric on out-of-distribution query types" lacks persuasiveness since it contradicts the results presented in Table 3.

- Regarding the comparison between BiQE and SQE+Transformer's performance in Section 5.4, I wonder if the transformer models used are of the same size. While the encoder part of BiQE employs a BERT model, SQE+Transformer uses three layers. If the sizes are the same, it would be helpful to have clarification on this point. However, if they are not, more experiments should be conducted to ensure a fairer comparison.

3. This paper lacks a more thorough analysis of the drawbacks of the proposed method, such as the performance on different types of queries or the scalability to larger KGs.

4. The writing can be improved. There are some typos and unclear descriptions. Please refer to the comments for detail.

---

> ### Author Response · Authors · 2023-04-16
> **Responses**
>
> Thanks for your review! We are happy to make the following clarifications.
>
> RE：W1
>
> In order to ensure fair comparisons between baselines and SQE models in the task of CQA, we use the same query embedding size for all models. This is a widely adopted convention in the field [1][2]. However, the total number of neural network parameters may still differ between baseline models. We provide the number of parameters for each baseline in the following table.
> To further ensure fairness, we set the number of layers to 3 in all SQE models, which ensures that the total number of parameters in SQE models is roughly the same or less than in baseline models. By doing so, we guarantee that comparisons between the two types of models are fair.
> In addition, evaluating Transformer and LSTM models with the same number of layers is a common practice when assessing their compositional generalization [3][4]. Therefore, we have adopted this experiment setting to ensure that our results can be compared fairly with previous studies.
>
> We updated the paper by including the table in the appendix of the paper, with corresponding clarification in section 4.4.
>
> | Models                 |                   | Number of Parameters |
> |------------------------|-------------------|----------------------|
> |       Baselines        | GQE               | 6.2 million          |
> |                        | Q2B               | 6.6 million          |
> |                        | MLP               | 7.6 million          |
> |                        | BetaE             | 13.7 million         |
> |                        | ConE              | 18.9 million         |
> | SQE Models  (3 layers) | SQE + CNN         | 7.1 million          |
> |                        | SQE + GRU         | 8.1 million          |
> |                        | SQE + LSTM        | 8.8 million          |
> |                        | SQE + Transformer | 14.7 million         |
>
> RE：W2
>
> Our main focus in this paper is on FOL queries, which involve projection, intersection, union, and negation. Table 4 presents the results for general FOL queries. Therefore, we can conclude that SQE "performs worse on Entailment than previous models" and "performs comparably to previous QE methods in the Inference metric on out-of-distribution query types" hold true for general FOL queries.
> Table 3 provides the results for conjunctive queries, which is a simpler subset of FOL queries that only includes projections and intersections (without unions and negations). Our experiments show that the SQE model consistently outperforms models that are specifically designed for encoding conjunctive queries. It's important to note that this observation is not contradictory to our conclusion on the general FOL queries. We provide further explanation for these performance differences in section 4.5.
> Regarding the comparison between BiQE and SQE+Transformer, we would like to clarify that the sizes of the SQE+Transformer and BiQE models in our experiments are the same. It is also worth noting that BiQE does not use BERT. Instead, it adopts the Transformer encoder structure and does not use the pre-trained weights of BERT.
> We have updated the related information in section 5.4.
>
> RE：W3
>
> We have added two new tables, Table 12 and 13,  to analyze queries of different depths. Our analysis shows that the performance of queries of various depths is consistent with the results reported in Table 4. Specifically, SQE is better at encoding queries whose types were observed during the training steps. We also acknowledge the weakness of SQE in encoding out-of-distribution queries in the entailment metric in the paper. Furthermore, we found that SQE does not suffer from any scalability issues when compared to previous methods.
>
> RE：W4
>
> Thank you for pointing this out, and we have fixed them accordingly.
>
>
> RE: RC 1
>
> Initially, we disregarded the results of GRU and CNN in Table 3 to avoid redundancy and leave space for more important discussions in the following sections of the paper. We believed that the results were redundant since we showed in Table 4 that LSTM and Transformer generally outperformed GRU and CNN on FOL queries. However, we understand your concerns and have listed the results of GRU and CNN in Table 3 in the revised version of the paper.
>
> RE: RC 2
>
> We have added some discussions in the related work section regarding the QTO paper. QTO is a concurrent work to our paper, and while it is also a method proposed for solving CQA, it is not a query encoding method. Instead, QTO leverages pre-trained graph embeddings and proposes an effective way to search in the latent space. QTO achieves high performance on CQA at the cost of high inference complexity. The time complexity of QTO on a query of size L and vocabulary size V is O(V^2 * L), which can be challenging to scale up. In contrast, the time complexity of SQE+LSTM is O(L), and SQE+Transformer is O(L^2). We have added a discussion of the QTO paper in Section 6 of our paper.

---

> > ### Author Response · Authors · 2023-04-16
> > **Reference**
> >
> > [1] Hongyu Ren, Weihua Hu, and Jure Leskovec. Query2box: Reasoning over knowledge graphs in vector space using box embeddings. In ICLR, 2020.
> > [2] Hongyu Ren and Jure Leskovec. Beta embeddings for multi-hop logical reasoning in knowledge graphs. In Advances in Neural Information Processing Systems, volume 33, 2020.
> > [3] Najoung Kim and Tal Linzen. 2020. COGS: A Compositional Generalization Challenge Based on Semantic Interpretation. In Proceedings of EMNLP-2020
> > [4] Wu, Zhengxuan, Christopher D. Manning, and Christopher Potts. "ReCOGS: How Incidental Details of a Logical Form Overshadow an Evaluation of Semantic Interpretation." arXiv preprint arXiv:2303.13716 (2023).

---

> > ### Comment · Reviewer_u3jx · 2023-04-21
> > **Thanks for the response**
> >
> > Thanks for the response. My concerns have been addressed.

---

### Review · Reviewer_xVu5 · 2023-04-02

**Summary Of Contributions:**

In this work, the authors present Sequential query encoding (SQE), as an alternative to traditional query encoding methods for Complex Query Answering (CQA) in knowledge graph reasoning. SQE uses a search-based algorithm to linearize the computational graph to a sequence of tokens and then uses a sequence encoder to compute its representation, which is used as a query embedding to retrieve answers from the embedding space according to similarity scores. Despite its simplicity, SQE demonstrates state-of-the-art performance on an extended benchmark including in-distribution queries and comparable knowledge inference capability on out-of-distribution queries.

**Audience:**

Yes

**Claims And Evidence:**

Yes

**Requested Changes:**

- It would be better if the results of CQD-CO, FuzzQE on the given benchmark were reported.
- More analyses on the out-of-distribution queries are expected. e.g., why SQE does not bring benefits. It seems that Tree LSTM is promising for out-of-distribution queries.


**Strengths And Weaknesses:**

Strengths:
- The proposed method is novel. As the paper claims, it is the first method that uses sequence encoders to encode linearized computational graphs for CQA on knowledge graph reasoning.
- A new evaluation set is created, and the performance of the proposed approach is promising.
- The finding that sequential encoding methods demonstrate better performance than computational graph based methods is insightful for future methodology design.
- The paper is well-written and easy to follow.

Weaknesses:
- Some recent state-of-the-art methods (CQD-CO, FuzzQE[a]) are missing in the comparison table.
- As shown in Table 4, SQE does not seem advantageous in out-of-distribution query modeling. The performance on out-of-distribution queries is an important indicator of the generalization capability of the algorithm.

[a]  Fuzzy Logic Based Logical Query Answering on Knowledge Graphs

---

> ### Author Response · Authors · 2023-04-16
> **Responses**
>
> Thank you for your review! We updated the paper with your suggestions and would like to make the following clarifications.
>
> RE: Requested Change 1
>
> We have added the FuzzQE implementation to our baseline methods. We trained and evaluated the FuzzQE and reported the results in Table 4 together with other baseline models. The conclusions of this paper are not affected by adding this baseline.
> We evaluated the performance of CQD, a search-based method that utilizes pre-trained link predictors for inference-time optimization, on our benchmark of conjunctive queries. To conduct this evaluation, we used the open-sourced code of CQD and their pre-trained link predictors and reported the results in Table 9. As CQD does not support the negation operator, we evaluated its performance on our benchmark of conjunctive queries. It should be noted that CQD is only trained on 1p queries, so we used the terms "in-distribution" and "out-of-distribution" to indicate the sets of query types evaluated. Our experimental results show that while the CQD model outperformed some baseline models, it was still unable to outperform SQE methods.
>
> RE: Requested Change 2
>
> Thank you for your suggestions. We have added a discussion of the differences between SQE+LSTM and the Tree-LSTM model in section 5.2.
>
> Both the LSTM and Tree-LSTM models use the same encoding unit of LSTM cells. However, in the LSTM model, the LSTM cells are sequentially connected, whereas in the Tree-LSTM model, the LSTM cells are connected from the leaves to the root following a tree structure.
>
> When using SQE+LSTM, the structural information of complex queries is encoded in the input tokens and is purely learned in the optimization process. However, for the Tree-LSTM model, the structural information, i.e., how to connect the LSTM cells, is directly given even for queries whose types are not observed during the training process. This can explain why Tree-LSTM performs better than SQE+LSTM on out-of-distribution queries. However, for in-distribution queries, the problem of query encoding is a pure sequence encoding task, and the LSTM model is already trained on how to leverage the structural information on the observed query types. Therefore, LSTM performs better than Tree-LSTM on in-distribution queries.

---

### Review · Reviewer_8u9c · 2023-04-06

**Summary Of Contributions:**

The paper proposed a method to linearize computation graphs into plain sequences and learn a Transformer model to encode the linearized query into an embedding. The query embedding is then used to retrieve answers using an inner-product search.

The authors experimented their model on three popular KB, FB15k, FB15k-237, and NELL. They evaluate their model on both entailment and inference tasks, for both in-distribution and out-of-distribution queries.

**Audience:**

Yes

**Broader Impact Concerns:**

The proposed method can be an elegant QE methods for compositional reasoning tasks if claims made in this paper can be properly justified.

**Claims And Evidence:**

No

**Requested Changes:**

Please see my questions above.

**Strengths And Weaknesses:**

Strengths:
1. The paper proposed an elegant way to encode compositional queries to perform reasoning tasks.
2. The authors conducted thorough ablation study to justify their design choice.

Weakness:
I don't observe any specific weakness of this paper. But I have some questions below for the authors.

A few questions:
1. It seems "SQE + Transformer" performs well on in-distribution queries, while "SQE + Transformer" performs well on out-of-distribution queries. Could you please provide some analysis on this? Is it because Transformer is better at overfitting the seen queries?

2. During training, models are provided with, e.g. FB15k, 273,710 queries with type p. Is this the entire FB15k KB? If so, is it likely that models have already memorized the KB? If models have already memorized the KB, why is it not fair to compare to (Sun et al., 2020; Xu et al., 2022; Zhu et al., 2022)? I agree that comparing to such models with the entailment task may be unfair, but one should consider comparing with these models with the inference tasks.

3. The linearized sequences are very similar to queries in the Q2B datasets, except for some minor changes in schema. (Please correct me if I am wrong). Is the proposed schema better for any particular reason?

4. One of the motivation is to avoid iterative encoding process commonly used in previous methods. Can you please compare with the baselines with varying number of reasoning steps?

5. In section 4.3, the authors mentioned that some of the baseline models do not support union and negation. Are numbers in Table 3 reported on the tasks include "union and negation"? If so, how do the numbers of the proposed method compare to the baselines on tasks without "union and negation"?

6. Why the new benchmark only contains 29 types of out-of-distribution queries? As motivated by the paper, there can be exponentially many types of queries and the queries can be automatically generated.

---

> ### Author Response · Authors · 2023-04-16
>
> Thank you so much for your detailed reviews! We try to answer those questions one by one.
>
> RE: Q1
>
> The poor performance of the SQE+transformer cannot be attributed solely to overfitting, as the validation scores of both in-distribution and out-of-distribution queries show an increasing trend until convergence. This indicates that there is no overfitting to in-distribution query types. In Section 5.4, we have provided some explanations for why SQE+transformer performs poorly on out-of-distribution queries. The reason could be that the existing positional encoding design in the transformer is not effective at utilizing unseen structural information in the input tokens.
>
> RE: Q2
>
> First, we would like to clarify that this paper does not claim SQE to be the state-of-the-art (SOTA) method for CQA. Instead, this paper claims that SQE is the SOTA neural query encoder for CQA, and this claim is well-justified by the extensive experiments conducted in this paper.
> Second, while we acknowledge the existence of neural-symbolic query encoders (Sun et al., 2020; Xu et al., 2022; Zhu et al., 2022), we did not compare them in this paper as their contribution is orthogonal to our focus on neural query encoding. It should be noted that SQE can be used to replace the neural query encoders used in these methods.
>
> RE: Q3
>
> The query2box datasets use query type names such as 1p, 2p, ip, pi, and 2u to represent query types. However, the rules behind these names are unknown, and they have limited scope. In this paper, we opted not to use these names and instead referred to the query types using lisp-like representations. This decision was made because we had 58 types of queries, and it was not feasible to use names to represent them. Additionally, we designed the schema for the SQE dataset slightly differently to ensure all sub-queries are enclosed by brackets, which better indicates the graph structures.
>
>  RE: Q4
>
> We have incorporated your suggestion and added two tables, Table 12 and 13, to demonstrate the performance of FOL queries with varying depths. The query depth is used to measure the maximum number of inference steps in the computational graph of a query. Our analysis confirms that the performance of queries with different depths is consistent with the results presented in Table 4. Specifically, SQE outperforms other encoders when encoding queries whose types were observed during the training process.
>
> RE: Q5
>
> In Table 3, we report the performance scores of the baseline models on conjunctive queries, which have logical expressions consisting only of conjunctions and no disjunction or negation operators. In Section 4.3, we demonstrate that some baseline models are unable to handle union and negation operations and therefore, we only evaluate them on conjunctive queries. We provide more detailed definitions to clarify this in section 2. Additionally, we included a table (Table 8) that shows the query structures of the conjunctive queries in more detail.
>
> RE: Q6
>
> We provided a detailed explanation of our benchmark construction in section 4.2. We ensured that the number of in-distribution and out-of-distribution query types for evaluation are equal, and we randomly selected 29 types of queries from a pool of 301 types of queries, with a maximum query depth of three and three anchor nodes. As shown in Table 4, our benchmark is already four times larger than previous benchmarks. With 58 evaluation types, we have a total of 2 million evaluation examples, which is sufficient for our problem settings. Therefore, evaluating more types of queries is not only impractical but also unnecessary.

---

### Decision · Action_Editors · 2023-05-29

**Recommendation:** Accept with minor revision

**Comment:**

Two of the major weaknesses of the paper are the lack of novelty (the method consists in linearizing a FOL query) and the breadth of the exploration, particularly the lack of references to the extensive amount of work that deals with compositional generalization in semantic parsing. For the latter, this can be a useful starting point: https://aclanthology.org/2022.acl-long.251.pdf.

Nevertheless, reviewers felt that the paper was improved after the rebuttal and the discussion with the authors. I think the paper can be interesting as it sets some useful baselines for this task. I choose to Accept this paper with minor revision. I would love for the authors to incorporate a reference to https://aclanthology.org/2022.acl-long.251.pdf, and hope that they will put additional work to incorporate some of the techniques in the paper to improve OOD scores of their SQE+Transformer baseline.


**Audience:**

Per se, the paper is not very novel. However, the study on how models perform in-distribution/out-of-distribution for CQA might be interesting to some readers.

**Claims And Evidence:**

This paper tackles a complex question answering task where queries are defined in first-order logical form. Differently from approaches that encode queries with architectures taking into consideration the query structure in their computation graph, this paper studies whether it is possible to embed the structure in a specific linearized form (i.e. as a sequence) and run a LSTM / Transformer encoder on top of the linearized structure. The results suggest that running a LSTM on top of the linearized query performs as well as established methods that take into account the query structure explicitly in the computation graph (TreeLSTM, etc..) and it performs on-par to these methods on out-of-distribution queries. Encoding the linearized query with a Transformer instead lead to worse out-of-distribution scores.